

# Identification of hydrological model parameters variation using
# ensemble Kalman filter
Chao Deng[1,2], Pan Liu[1,2,*], Shenglian Guo[1,2], Zejun Li[1,2], Dingbao Wang[3]
[1]State Key Laboratory of Water Resources and Hydropower Engineering Science, Wuhan University,
Wuhan 430072, China
[2]Hubei Provincial Collaborative Innovation Center for Water Resources Security, Wuhan 430072,
China
[3]Department of Civil, Environmental & Construction Engineering, University of Central Florida,
Orlando 32816, USA
*Corresponding author: P. Liu, State Key Laboratory of Water Resources and Hydropower
Engineering Science, Wuhan University, Wuhan 430072, China
Email: liupan@whu.edu.cn
Tel: +86-27-68775788; Fax: +86-27-68773568



**Abstract**: Hydrological model parameters play an important role in the ability of model prediction. In a stationary context, parameters of hydrological models are treated as constants; however, model parameters may vary with time under climate change and human activities. The technique of ensemble Kalman filter (EnKF) is proposed to identify the temporal variation of parameters for a two-parameter monthly water balance model (TWBM) by assimilating the runoff observations. Through a synthetic experiment, the proposed method is evaluated with time-invariant (i.e., constant) parameters and different types of parameter variations, including trend, abrupt change, and periodicity. Various levels of observation uncertainty are designed to examine the performance of the EnKF. The results show that the EnKF can successfully capture the temporal variations of the model parameters. The application to the Wudinghe basin shows that the water storage capacity ($SC$) of the TWBM model has an apparent increasing trend during the period from 1958 to 2000. The identified temporal variation of $SC$ is explained by land use and land cover changes due to soil and water conservation measures. Whereas, the application to the Tongtianhe basin shows that the estimated $SC$ has no significant variation during the simulation period of 1982-2013, corresponding to the relatively stationary catchment characteristics. The evapotranspiration parameter ($C$) has temporal variations while no obvious change patterns exist. The proposed method provides an effective tool for quantifying the temporal variations of the model parameters, thereby improving the accuracy and reliability of model simulations and forecasts.



**Keywords**: model parameter identification, temporal variation of parameter, catchment characteristics,

ensemble Kalman filter



# 1 Introduction

Hydrological model parameters are critically important for accurate simulation of runoff. Parameters of

conceptual hydrological models can be considered as a simplified representation of the physical

characteristics in hydrologic processes. Therefore, parameter values are closely related to the catchment

conditions, such as climate change, afforestation and urbanization (Peel et al., 2011). In hydrological

modeling, parameters are usually assumed to be stationary, i.e., the calibrated parameters are constants

during the calibration period, and have extrapolative ability outside the range of the observations used

for parameter estimation (Merz et al., 2011). However, the calibration period may contain different

climatic conditions and hydrological regimes compared to the simulation period (Merz et al., 2011;

Zhang et al., 2011; Westra et al., 2014; Patil and Stieglitz, 2015). The model parameters may change

responding to the variations in climatic conditions and catchment properties. For example, land use and

land cover changes contribute to temporal changes of model parameters (Andréassian et al., 2003;

Brown et al., 2005; Merz et al., 2011). As a result, the estimated parameters usually depend on the

calibration period (Merz et al., 2011; Coron et al., 2012; Seiller et al., 2012; Westra et al., 2014).

Therefore, assuming time-invariant model parameters may be unrealistic, especially for catchments with

time-varying climate conditions and/or catchment characteristics.

The time-variant hydrological model parameters has been reported in a few recent publications (Merz et



al., 2011; Brigode et al., 2013; Jeremiah et al., 2014; Thirel et al., 2014; Westra et al., 2014; Patil and
Stieglitz, 2015). For example, Ye et al. (1997) and Paik et al. (2005) mentioned the seasonal variations
of hydrological model parameters. Merz et al. (2011) analyzed the temporal changes of model
parameters, which were calibrated respectively by using six consecutive 5-year periods between 1976
and 2006 for 273 catchments in Austria. Recently, Westra et al. (2014) proposed a strategy to cope with
nonstationarity of hydrological model parameters, which were represented as a function of a
time-varying covariate set before using an optimization algorithm for calibration. Previous studies
provided two main methods to estimate the time-variant model parameters: (1) Parameters are estimated
for each consecutive subsets divided from the historical record using an optimization algorithm (Merz et
al., 2011; Thirel et al., 2015); (2) A functional form of the selected time-variant model parameters is
constructed and, the parameters for the function are estimated using an optimization algorithm based on
the entire historical record (Jeremiah et al., 2014; Westra et al., 2014).

The data assimilation (DA) actually provides another method to identify the potential temporal
variations of model parameters by updating them in real-time when observations are available (Liu and
Gupta, 2007; Xie and Zhang, 2013). The DA method has been widely applied in hydrology for soil
moisture estimation (Han et al., 2012; Kumar et al., 2012) and flood forecasting (Liu et al., 2013; Abaza
et al., 2014). It has also been successfully used to estimate model parameters (Moradkhani et al., 2005;



Panzeri et al., 2013; Vrugt et al., 2013; Xie and Zhang, 2013; Shi et al., 2014; Xie et al, 2014). For
example, Vrugt et al. (2013) proposed two types of Particle-DREAM method to track the evolving
target distribution of HyMOD parameters, while the true parameters were assumed to be constant. Xie
and Zhang (2013) used a partitioned forecast-update scheme based on the EnKF to retrive optimal
parameters in a distributed hydrological model. Although the DA method has been used to estimate
model parameters, these studies are focused on the estimation of constant parameters. Little attention
has been paid to the identification of time-variant model parameters and the interpretation of their
temporal variations based on the climate conditions and/or catchment characteristics.

The aim of this study is to assess the capability of the DA method (i.e., the EnKF) to identify the
temporal variations of the model parameters for a monthly water balance model. Thus, a synthetic
experiment, including four scenarios with different parameter variations and one scenario with
time-invariant parameters, is designed for parameter estimation at different uncertainty levels.
Furthermore, two case studies are implemented to estimate the model parameter series and to interpret
the parameter variations in response to the changes in catchment characteristics, i.e., land use and land
cover. The remainder of this paper is organized as follows. Section 2 presents a brief review of the
monthly water balance model and the EnKF method. Following the methodology, Section 3 describes
the synthetic experiment and the application to two case studies. Results and discussion are presented in





Section 4, followed by conclusions in Section 5.

## 2 Methodology

### 2.1 Monthly water balance model

The two-parameter monthly water balance model (TWBM), developed by Xiong and Guo (1999), has

been widely applied for monthly runoff simulation and forecast (Guo et al., 2002; Guo et al., 2005;

Xiong and Guo, 2012; Li et al., 2013; Zhang et al., 2013; Xiong et al., 2014). The inputs of the model

include monthly areal precipitation and potential evapotranspiration. The actual monthly

evapotranspiration is calculated as follows:

$$E_i = C \times EP_i \times \tanh\left(P_i / EP_i\right) \tag{1}$$

where $E_i$ represents the actual monthly evapotranspiration; $EP_i$ and $P_i$ are the monthly potential

evapotranspiration and precipitation, respectively; $C$ is the first model parameter; and $i$ is the time

step.

The monthly runoff is dependent on the soil water content and is calculated by the following equation:

$$Q_i = S_i \times \tanh\left(S_i / SC\right) \tag{2}$$

where $Q_i$ is the monthly runoff; and $S_i$ is the soil water content. As the second model parameter,

$SC$ represents the water storage capacity of the catchment in millimeter. The available water for





runoff at the $i$ th month is computed by $S_{i-1} + P_i - E_i$. Then, the monthly runoff is calculated as:
$$Q_i = \left(S_{i-1} + P_i - E_i\right) \times \tanh\left[\left(S_{i-1} + P_i - E_i\right)/SC\right]$$   (3)

Finally, the soil water content at the end of each time step is updated based on the water conservation
law:
$$S_i = S_{i-1} + P_i - E_i - Q_i$$   (4)

## 2.2 Ensemble Kalman filter

EnKF is a typical sequential data assimilation technique based on the Monte Carlo method and
produces an ensemble of state simulations to update the state variables and model parameters,
conditioned on a series of observations (Moradkhani et al., 2005; Shi et al., 2014). It is applicable to a
variety of nonlinear problems (Evensen, 2003; Weerts and El Serafy, 2006) and has been widely
applied to hydrological models (Abaza et al., 2014; DeChant and Moradkhani, 2014; Delijani et al.,
2014; Samuel et al., 2014; Tamura et al., 2014; Xue and Zhang, 2014; Deng et al., 2015). Furthermore,
the EnKF has been successfully used in time-invariant parameter estimations for hydrological models
(Moradkhani et al., 2005; Wang et al., 2009; Xie and Zhang, 2010; Xie and Zhang, 2013).

In this paper, the EnKF is applied to simultaneously estimate state variables and parameters (**Table 1**)




in the TWBM model. The augmented state vector includes both states and model parameters (Wang et
al., 2009), i.e., $Z = \begin{pmatrix} \theta \\ x \end{pmatrix}$, where $\theta$ includes the evapotranspiration parameter $C$ and the catchment
water storage capacity $SC$, and $x$ is the soil water content $S$. The model forecast is conducted for
each ensemble member as follows:
$$\begin{pmatrix} \theta_{i+1|i}^k \\ x_{i+1|i}^k \end{pmatrix} = \begin{pmatrix} \theta_{i|i}^k \\ f\left(x_{i|i}^k, \theta_{i+1|i}^k, u_{i+1}\right) \end{pmatrix} + \begin{pmatrix} \delta_i^k \\ \varepsilon_i^k \end{pmatrix}, where \ \delta_i^k \sim N\left(0, U_i\right), \varepsilon_i^k \sim N\left(0, G_i\right) \tag{5}$$

where $\theta_{i+1|i}^k$ is the $k$th ensemble member forecast of model parameters at time $i+1$; $\theta_{i|i}^k$ is the $k$th
updated ensemble member of model parameters at time $i$; $x_{i+1|i}^k$ is the $k$th ensemble member forecast
of model state at time $i+1$; $x_{i|i}^k$ is the $k$th updated ensemble member of model state at time $i$; $f$ is
the forecasting model operator, i.e. the TWBM model; $u_{i+1}$ is the forcing data for the hydrological
model, including precipitation and potential evapotranspiration; $\varepsilon_i^k$ and $\delta_i^k$ are the independent
white noise for the forecasting model, followed a Gaussian distribution with zero mean and specified
covariance $G_i$ and $U_i$, respectively. Note that the parameters in Eq. (5) are propagated by adding
random disturbances to the parameter member between time steps (Wang et al., 2009).

The observation ensemble member can be written as:
$y_{i+1}^k = h\left(x_{i+1|i}^k, \theta_{i+1|i}^k\right) + \xi_{i+1}^k, \ \xi_{i+1}^k \sim N\left(0, W_{i+1}\right)$ (6)
where $y_{i+1}^k$ is the $k$th ensemble member of the model simulated runoff at time $i+1$; $h$ is the





observation operator which represents the relationship between the observation and the state variables;
$\xi_{i+1}^k$ is the noise term which follows a Gaussian distribution with zero mean and specified covariance
$W_{i+1}$.

Based on the available state and observation equations, the model parameters and state are updated
according to the following equation:
$$Z_{i+1|i+1}^k = Z_{i+1|i}^k + K_{i+1}\left( y_{i+1}^k - h\left( Z_{i|i}^k \right) \right) \tag{7}$$
where $Z$ is the augmented state vector that includes both state and parameters; $y_{i+1}^k$ is the $k$th
observation ensemble member generated by adding the observation error $\xi_{i+1}^k$ to the observed runoff:
$$y_{i+1}^k = y_{i+1} + \xi_{i+1}^k \tag{8}$$
$K_{i+1}$ is the Kalman gain matrix that represents the weight between the forecasts and observations. It
can be calculated as (Moradkhani et al., 2005):
$$K_{i+1} = \Sigma_{i+1|i}^{zy}\left( \Sigma_{i+1|i}^{yy} + W_{i+1} \right)^{-1} \tag{9}$$
where $\Sigma_{i+1|i}^{zy}$ is the cross covariance of the forecasted state and parameters; $\Sigma_{i+1|i}^{yy}$ is the error
covariance of the forecasted output. The error covariance matrix is calculated based on the forecasted
ensemble members:
$$\Sigma_{i+1|i} = \frac{1}{N-1} Z_{i+1|i} Z_{i+1|i}^T \tag{10}$$
where $Z_{i+1|i} = \left( z_{i+1|i}^1 - z_{i+1|i}^m, \cdots, z_{i+1|i}^N - z_{i+1|i}^m \right)$ and $z_{i+1|i}^m$ is the ensemble mean of the forecasted members,





and  $N$  is the ensemble size.

Since the parameters are limited within a range, the constrained EnKF (Wang et al., 2009) is used in this
study. The ensemble size, uncertainties in input and output have significant impacts on the assimilation
performance of the EnKF, and they are specified following the previous studies (Moradkhani et al.,
2005; Wang et al., 2009; Xie and Zhang, 2010; Nie et al., 2011; Lü et al., 2013; Samuel et al., 2014).
Generally, larger ensemble size causes the propagation of more accurate error information but leads to
computational burden (Moradkhani et al., 2005; Xie and Zhang, 2010). In this study, there are only
three variables including two model parameters and one state variable in the assimilation process. To
satisfy the estimation accuracy and the computational efficiency, the ensemble size is set to 1000 for the
synthetic experiment and the two case studies. In the present study, the uncertainties, including state
variable and parameter errors ( $\varepsilon$  and  $\delta$ in Eq. (5), respectively), and runoff observation error ( $\xi$  in Eq.
(6)), are assumed to follow a Gaussian distribution with zero mean and specified covariance. Note that
the model parameter errors should vary relying on the hydrological model used and the study basin
(Clark et al., 2008). Larger standard deviation can generate greater perturbations to model parameters,
and it can improve the coverage of updated parameters but also may cause fluctuations in the estimates.
In this study, the parameter errors are determined empirically, i.e., the standard deviation of  $C$  is set to
0.01 for all the cases, while that of  $SC$  is set to 5.0, 1.0 and 0.5 in the synthetic experiment, Wudinghe





basin and Tongtianhe basin, respectively. The standard deviations of both model state and observation
errors are assumed to be proportional to the magnitude of true values (Wang et al., 2009; Lü et al.,
2013). The proportional factors of model state are set to 0.05 for all the cases. Different proportional
factors of runoff observation and precipitation (**Table 3**) are evaluated to examine the capability of the
EnKF in the synthetic experiment; whereas, the proportional factors of runoff observation are set to 0.1
and zero precipitation errors are assumed in the two case studies. It should be noted that the variable
variance multiplier can be used to perturb the observations (Leisenring and Moradkhani, 2012; Yan et
al., 2015).

## 2.3 Evaluation index
Two evaluation criteria, including the Nash-Sutcliffe efficiency (*NSE*) (Nash and Sutcliffe, 1970) and
the volume error (*VE*) are used to evaluate the runoff assimilation results for the synthetic experiment
and the application to real catchments (Deng et al., 2015; Li et al., 2015).
$$NSE = 1 - \frac{\sum_{i=1}^{n}\left(Q_{sim,i} - Q_{obs,i}\right)^2}{\sum_{i=1}^{n}\left(Q_{obs,i} - \overline{Q}_{obs}\right)^2} \tag{11}$$
$$VE = \frac{\sum_{i=1}^{n} Q_{sim,i} - \sum_{i=1}^{n} Q_{obs,i}}{\sum_{i=1}^{n} Q_{obs,i}} \tag{12}$$
where $Q_{sim,i}$ and $Q_{obs,i}$ are the simulated and observed runoff for the $i$th month; $\overline{Q}_{obs}$ is the mean
values of the observed runoff; and $n$ is the total number of data points. The *NSE* has been widely




used to assess the goodness-of-fit for hydrological modeling. A *NSE* value of 1 means a perfect match
of simulated runoff to the observations. The *VE* is a measure of bias between the simulated and
observed runoff. For example, *VE* with the value of 0 denotes no bias, and a negative value means an
underestimation of the total runoff volume.

The assimilated parameter results are evaluated using the following criteria, including the Pearson
correlation coefficient (*R*), the root mean square error (*RMSE*) and mean absolute relative error
(*MARE*):
$$R = \frac{\sum_{i=1}^{n}\left(\theta_{sim,i} - \bar{\theta}_{sim}\right)\left(\theta_{obs,i} - \bar{\theta}_{obs}\right)}{\sqrt{\sum_{i=1}^{n}\left(\theta_{sim,i} - \bar{\theta}_{sim}\right)^2 \left(\theta_{obs,i} - \bar{\theta}_{obs}\right)^2}}$$  (13)
$$RMSE = \sqrt{\frac{1}{n}\sum_{i=1}^{n}\left(\theta_{sim,i} - \theta_{obs,i}\right)^2}$$  (14)
$$MARE = \frac{1}{n}\sum_{i=1}^{n}\frac{\left|\theta_{sim,i} - \theta_{obs,i}\right|}{\theta_{obs,i}}$$  (15)
where $\theta_{sim,i}$ and $\theta_{obs,i}$ are the assimilated and true model parameters for the $i$ th month; $\bar{\theta}_{sim}$ and
$\bar{\theta}_{obs}$ are the mean of the assimilated and true model parameters, respectively for the $i$ th month; $n$ is
the total number of data points.

**3 Data and study area**





## 3.1 Synthetic experiment

A synthetic experiment is designed to evaluate the capability of the assimilation procedure to identify

the temporal variation of model parameters. Five scenarios of different parameter variations are

developed, as shown in **Table 2**. The model parameters in the first four scenarios are time-variant, and

those in the last scenario are constant. Parameter *C*, the evapotranspiration parameter, is considered to

be sinusoidal reflecting potential seasonal variations in hydrological model parameters (Paik et al., 2005;

Ye et al., 1997). An increasing trend is also considered to account for the potential annual or long-term

variability. The change of parameter *SC* is considered to be gradual and abrupt, since the catchment

water storage capacity can be affected by land use and land cover changes, such as afforestation and

dam construction. The parameters in Scenario 5 are treated as constants like the conventional

hydrological modeling. Observations for precipitation and potential evapotranspiration are generated by

adding a Gaussian disturbance to the corresponding data from a real catchment, and runoff is then

produced using the TWBM model. The data set used in this experiment includes a total of 672 months.

The first 24-month period is set for model warm-up to reduce the impact of the initial soil moisture

conditions. The steps toward identifying temporal variation of model parameters are as follows:

(1) Time series of model parameters are generated, including the time-variant parameters and the

constant parameters. Model parameter sets are produced using a sinusoidal function and/or a linear

trend function within the specified ranges shown in **Table 1**. The runoff observations for each scenario



are computed from the TWBM model taking monthly potential evapotranspiration and precipitation,
and the parameters as inputs.
(2) The initial ensembles of model parameters and state variables are generated using uniform
distributions within the specified ranges in **Table 1**. The ensemble size and the total number of
assimilation time steps are specified.
(3) After the initialization of parameters and state variables, the hydrological model parameters and
states are updated by assimilating the runoff observations obtained in Step (1). The additive errors for
generating the ensemble members of model parameters, state variables and runoff observations are
obtained from Gaussian distributions with zero mean and specified variance.

To evaluate the effect of errors on identifying parameter variation, different levels of observation
uncertainty are considered in the synthetic experiment, as detailed in **Table 3**. The uncertainties from
the observed precipitation and runoff are characterized by adding Gaussian noises where the standard
deviations are assumed to be proportional to the magnitude of the true values, and the corresponding
proportional factors are denoted as $\gamma_P$ and $\gamma_Q$. The proportional factors are set to account for the
practical measurement error (Wang et al., 2009; Xie and Zhang, 2010).

**3.2 Study area**



### 3.2.1 Case 1: Wudinghe basin

The method is applied to the Wudinghe basin (**Fig. 1**), which is a sub-basin of the Yellow River basin

and located in the southern fringe of Maowusu Desert and the northern part of the Loess Plateau in

China with a semiarid climate. It has a drainage area of approximately 30,261 km$^2$ and a total length of

491 km. The Wudinghe basin has an average slope of 0.2%, and its elevation ranges from 600 to 1800

m above the sea level. The Baijiachuan gauge station, which is the most downstream station of the

Wudinghe basin, drains 98% of the total basin area. The mean annual precipitation over the basin is

401 mm, of which 72.5% occurs in the rainy season from June to September (**Fig. 2**). The mean

annual potential evapotranspiration is 1077 mm, and the mean annual runoff is about 39 mm with a

runoff coefficient of 0.1.

The soil erosion is severe in the Wudinghe basin owing to the highly erodible loess and sparse

vegetation. Since the 1960s, the soil and water conservation measures have been undertaken. Lots of

engineering measures including tree and grass plantation, check dam and reservoir construction, and

land terracing were effectively implemented during several decades. The land use changes caused by

the soil and water conservation measures had a significant effect on increasing water storage capacity

(Xu, 2011).



### 3.2.2 Case 2: Tongtianhe basin

The Tongtianhe basin (**Fig. 3**) is located in southwestern Qinghai Province in China with a continental

climate. It belongs to the source area of Yangtze River basin with a drainage area of about 140,000 km$^2$

and a total main stream length of 1206 km. The elevation of the Tongtianhe basin approximately ranges

from 3500 to 6500 m above the sea level. Zhimenda is the basin outlet. The mean annual precipitation

over the basin is 440 mm, of which 76.9% occurs in the period from June to September (**Fig. 4**). The

mean annual potential evapotranspiration is 796 mm, and the mean annual runoff is about 99 mm with a

runoff coefficient of 0.23. The Tongtianhe basin is barely affected by human activities owing to the

limitation of the topographic condition and the water source protection guidelines conducted by the

government. The Tongtianhe basin is used for comparison on model parameter identification.

### 3.2.3 Data

The data sets used in this study include monthly precipitation, potential evapotranspiration and runoff in

Wudinghe basin (from 1956 to 2000) and Tongtianhe basin (from 1980 to 2013). The potential

evapotranspiration is estimated using the Penman-Monteith equation (Allen et al., 1998) based on the

meteorological data from the China Meteorological Data Sharing Service System (http://cdc.nmic.cn).

To reduce the impact of the initial conditions, a 2-year data set, i.e., from 1956 to 1957 for Wudinghe

basin and from 1980 to 1981 for Tongtianhe basin, is reserved as the warm-up period.



# 4 Results and discussion

## 4.1 Synthetic experiment

The comparisons of the estimated and true model parameters under different scenarios are presented

in **Fig. 3**, **Fig. 4** and **Fig. 5**. **Tables 4** and **5** show the evaluation statistics for the parameters and runoff

estimations. The assimilated parameter values are obtained from the ensemble mean at each time step.

The estimations of parameter $C$ and $SC$ have the similar trends as the true parameter series. The

temporal variations of the estimated $C$ agree well with the true series, although it has biases on the

peaks of the periodic changes. For $SC$, the temporal estimates can capture the different changes in

**Table 2**, especially for the abrupt change where the estimated values respond immediately. Different

uncertainty levels are considered to examine the capability of the EnKF method. The results in **Fig. 3**

show that the estimated $C$ has more accurate peaks with smaller *RMSE* and higher *R* values under the

high level uncertainty (**Table 4**); whereas, the $SC$ estimates in **Fig. 4** have some fluctuations when the

uncertainty level increases. This is due to the reason that the estimated values vary with increasing

uncertainty level in the assimilation process. In the synthetic experiment, the true $C$ is assumed to be

periodic with higher degree of variation, while the true $SC$ series have less variation.


It should be noted that there are time lags between the assimilated and true $C$. The observation at the



current time step is used to adjust the state variables and parameters in EnKF, and the updates of
parameters depend on the Kalman gain for parameters. A runoff observation at the current time is
determined by states at the current and previous time steps (Pauwels and Lannoy, 2006). The Kalman
gain is dependent on the relative value of observation error to model error. The updated states are
closer to the observation with a higher Kalman gain (Tamura et al., 2014). The synthetic $C$ series were
assumed to be periodic where lots of peak values exist; while the variation of $SC$ series is less. The
time lag between assimilated and true values exists especially when peak values occur (Clark et al.,
2008; Samuel et al., 2014).

The results for the scenario of constant parameters are shown in **Fig. 5**, demonstrating that the
estimated parameters can approach their true values after the initial 24 assimilation steps. The grey
areas represent the 95% uncertainty intervals, which reduce quickly and approach to a stable spread.
The performance of the estimated parameters is correlated with the uncertainty level. Higher
precipitation and runoff observation errors correspond to greater *RMSE* values (**Table 4**) of estimated
parameters and uncertainty ranges. The performance of runoff estimations for various parameter
changes under different levels of uncertainty is shown in **Table 5**, suggesting that the EnKF perfectly
matches the observations with NSEs higher than 0.95 and absolute VEs smaller than 0.02. The EnKF
can successfully capture the temporal variations of the true parameters, although the uncertainty levels



of the observations can affect its performance on a certain degree. The above results demonstrate that
the EnKF is able to identify the temporal variation of the model parameters by updating the state
variables and parameters based on the runoff observations.

## 4.2 Case studies

**Fig. 6** shows the double mass curve between monthly runoff and precipitation for the Wudinghe and
Tongtianhe basins, respectively. The top panel shows the linear relationship between cumulative runoff
and precipitation pre- and post-1972 in the Wudinghe basin, which is similar to the result presented by
Xu (2011) and Li et al. (2014). The results show two straight lines with different slopes for the
relationships between precipitation and runoff, indicating that an abrupt change occurred in 1972,
namely, the runoff generation had been changed from this year due to the soil and water conservation
measures. While the bottom panel demonstrates a single linear relationship fits all the data for the
Tongtianhe basin, suggesting a stable precipitation-runoff relationship during the 1982-2013 period.

The estimated parameters and the associated 95% uncertainty intervals are shown in **Fig. 7**. The time
series of estimated *SC* shows an apparent increasing trend, with two different trends for pre- and
post-turning point in **Fig. 6(a)**. The temporal variation of the water storage capacity is correlated with
the changes of land use and land cover. Both the trends in **Fig. 7(c)** show an increase of *SC*, because the




implementation of the large-scale engineering measures significantly improved the water holding
capacity of the Wudinghe basin, especially for the reservoir and check dam construction. The trend
slopes of the two periods, one is from 1956 to 1971, the other is from 1972 to 2000, are different
because the degree of implementing engineering measures varied during the period of 1958-2000.
Moreover, the increase of the water holding capacity slowed down during the 1980s due to the
sedimentation in reservoirs and check dams after periods of operation (Wang and Fan, 2003). **Fig. 8**
shows the runoff reduction caused by all the soil and water conservation measures, i.e., land terracing,
tree and grass plantation, check dam and reservoir construction. The runoff reduction positively relates
to the water holding capacity, namely the $SC$ value. The slope for the period of 1958-1971 is higher than
that for the period of 1972-1996, suggesting that the $SC$ in the former period has higher increasing trend.
The runoff reduction data is available from 1956 to 1996 (Wang and Fan, 2003). On the other hand, the
result of Tongtianhe basin shows that the estimated $SC$ has no detectable trend since the $R$ value has
an insignificance level. Moreover, the ranges and standard deviation of the estimated $SC$ values are
much smaller than those in the Wudinghe basin (**Fig. 7**), suggesting that the estimated $SC$ has no
obvious temporal variations.

For parameter $C$, the results show that the estimates have no obvious temporal patterns because the
trend line slopes are almost zero and the standard deviations are relatively small for the two basins (**Fig.**





**7(a)** and **(b)**). However, the temporal variations exist in the estimated $C$ values, indicating that this
parameter has different values during the time steps and can be treated as time-variant parameters. The
temporal variations of the estimated $C$ are related to the variation of monthly actual evaporation, which
is affected by multiple climatic factors, such as air temperature, soil moisture and solar irradiance (Su et
al., 2015). The grey regions represent the 95% uncertainty intervals obtained from the parameter
ensembles. The stable and narrow uncertainty bounds shown in **Fig. 7** indicate that the EnKF can
provide superior performance of parameter estimation. The runoff simulations for both the two basins
have good match with the runoff observations. Specifically, the *NSE* and *VE* for the Wudinghe basin are
0.93 and 0.07 respectively. While the corresponding index values are 0.99 and 0.04 for the Tongtianhe
basin.

In summary, the above results demonstrate that the EnKF can identify the temporal variation of model
parameters well by updating both state variables and parameters based on the runoff observations. The
trends of parameter  $SC$  can be explained by the changes of catchment characteristics (i.e., land use
and land cover) in the Wudinghe basin. However, the estimated  $SC$  for the Tongtianhe basin is
approximately stable with small standard deviation because the basin is located in a water protection
zone and has no significant changes on water storage capacity caused by human activities. The
parameter $C$ has temporal variations and can be treated as a time-variant parameter for both basins,



although the estimates have no obvious temporal patterns. Therefore, the EnKF is capable of identifying
the temporal variations of model parameters.

## 5 Conclusions

This study proposes an ensemble Kalman filter (EnKF) to identify the temporal variation of model
parameters in the two-parameter monthly water balance model (TWBM) by assimilating the runoff
observations. A synthetic experiment, which contains four scenarios with different changes of model
parameters and one scenario with constant parameters, is designed to examine the capability of the
proposed approach. Furthermore, three different levels of observation uncertainty are taken to assess
the performance of the EnKF. The main conclusions are drawn as follows: For the time-variant
parameters, the EnKF can provide superior performance even though slight time lags exist when
parameters have periodic variations. The true values of the constant parameters can be approached
quickly after 24 time steps of assimilation process. The temporal variations of the parameters can be
successfully captured under a high level of uncertainty, although the observation uncertainties from
precipitation and runoff have an influence on the performance of the EnKF.

The EnKF method is applied to the Wudinghe basin in China, aiming to detect the temporal variations
of the model parameters and to provide an explanation for the parameter variation from the perspective



of the catchment characteristic changes. Meanwhile, a comparison is implemented to investigate the
variation of model parameters in the Tongtianhe basin, which is barely affected by human activities. The
parameter of water storage capacity ($SC$) for the monthly water balance model shows a significant
increasing trend for the period of 1958-2000 in the Wudinghe basin. The soil and water conservation
measures, including land terracing, tree and grass plantation, check dam and reservoir construction,
have been implemented during 1958 to 2000, resulting in the increase of the water holding capacity of
the basin, which explains the increasing trends of $SC$. Moreover, the magnitudes of the engineering
measures in different time periods play an important role in the degree of increasing trend for $SC$. In the
Tongtianhe basin, the parameter $SC$ has no significant trend for the period of 1982-2013, which is
consistent with the relatively stationary catchment characteristics. The evapotranspiration parameter ($C$)
has temporal variations and can be treated as time-variant parameter, but no obvious trends exist.

The method proposed in this paper provides an effective tool for the time-variant model parameters
identification. Future work will be focused on the influence of the correlations between/among model
parameters and performance comparison of multiple data assimilation methods.

## Acknowledgments

This study was supported by the Excellent Young Scientist Foundation of NSFC (51422907) and the





Open Foundation of State Key Laboratory of Water Resources and Hydropower Engineering Science in
Wuhan University (2015SWG01). The authors would like to thank the editor and the anonymous
reviewers for their comments that helped to improve the quality of the paper.

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






# Tables

**Table 1.** States and parameters of the two-parameter monthly water balance model.

| Parameters and state variables | | Description | Ranges and unit |
|---|---|---|---|
| Parameter | $C$ | Evapotranspiration parameter | 0.2-2.0 (-) |
| | $SC$ | Catchment water storage capacity | 100-4000 (mm) |
| State variable | $S$ | Soil water content | mm |





**Table 2.** Different variations of model parameters in the synthetic experiment.

| Scenario | Description |
|---|---|
| Scenario 1 | $C$ has a periodic variation, and $SC$ has an increasing trend |
| Scenario 2 | $C$ has a periodic variation, and $SC$ has an abrupt change |
| Scenario 3 | $C$ has a periodic variation with an increasing trend, and $SC$ has an increasing trend |
| Scenario 4 | $C$ has a periodic variation with an increasing trend, and $SC$ has an abrupt change |
| Scenario 5 | Both $C$ and $SC$ are constant |







Table 3. Proportional factors of the standard deviations for precipitation ($\gamma_P$) and runoff ($\gamma_Q$) uncertainties.

| Type | Low level | Medium level | High level |
| --- | --- | --- | --- |
| $\gamma_P$ | 0 | 0.05 | 0.10 |
| $\gamma_Q$ | 0.05 | 0.10 | 0.20 |





**Table 4.** Performance statistics for various changes of (a) parameter *C* and (b) *SC* estimations under different levels of uncertainty in the synthetic experiment.

| Scenario | Low level | | | Medium level | | | High level | | |
|---|---|---|---|---|---|---|---|---|---|
| | *RMSE* | *MARE* | *R* | *RMSE* | *MARE* | *R* | *RMSE* | *MARE* | *R* |
| (a) Parameter *C* | | | | | | | | | |
| Scenario 1 | 0.15 | 0.21 | 0.55 | 0.16 | 0.18 | 0.68 | 0.18 | 0.11 | 0.89 |
| Scenario 2 | 0.16 | 0.19 | 0.63 | 0.17 | 0.16 | 0.75 | 0.18 | 0.09 | 0.91 |
| Scenario 3 | 0.12 | 0.13 | 0.64 | 0.13 | 0.11 | 0.72 | 0.14 | 0.07 | 0.91 |
| Scenario 4 | 0.13 | 0.12 | 0.70 | 0.13 | 0.10 | 0.77 | 0.14 | 0.06 | 0.93 |
| Scenario 5 | 0 | -- | -- | 0 | -- | -- | 0 | -- | -- |
| (b) Parameter *SC* | | | | | | | | | |
| Scenario 1 | 182.87 | 0.03 | 0.99 | 187.76 | 0.05 | 0.94 | 253.35 | 0.83 | 0.83 |
| Scenario 2 | 158.30 | 0.04 | 0.96 | 167.47 | 0.07 | 0.91 | 189.59 | 0.80 | 0.80 |
| Scenario 3 | 180.20 | 0.03 | 0.99 | 183.06 | 0.04 | 0.97 | 215.04 | 0.88 | 0.88 |
| Scenario 4 | 156.42 | 0.03 | 0.97 | 158.50 | 0.05 | 0.93 | 170.90 | 0.86 | 0.86 |
| Scenario 5 | 1.54 | -- | -- | 3.67 | -- | -- | 20.54 | -- | -- |





Table 5. Performance of runoff estimations for various parameter changes under different levels of uncertainty in the
synthetic experiment.

| Scenario | Low level | | Medium level | | High level | |
|---|---|---|---|---|---|---|
| | *NSE* | *VE* | *NSE* | *VE* | *NSE* | *VE* |
| Scenario 1 | 0.999 | -0.0003 | 0.988 | -0.0046 | 0.967 | -0.0230 |
| Scenario 2 | 0.999 | 0.0001 | 0.990 | -0.0028 | 0.967 | -0.0141 |
| Scenario 3 | 0.999 | -0.0011 | 0.990 | -0.0013 | 0.974 | -0.0264 |
| Scenario 4 | 0.999 | -0.0009 | 0.992 | 0.0002 | 0.959 | -0.0147 |
| Scenario 5 | 0.999 | -0.0022 | 0.992 | -0.0077 | 0.961 | -0.0187 |





**Figures**
**Figure. 1.** Location and mean monthly precipitation and runoff from 1956 to 2000 of the Wudinghe basin.






**Figure. 2.** Location and mean monthly precipitation and runoff from 1980 to 2013 of the Tongtianhe basin.




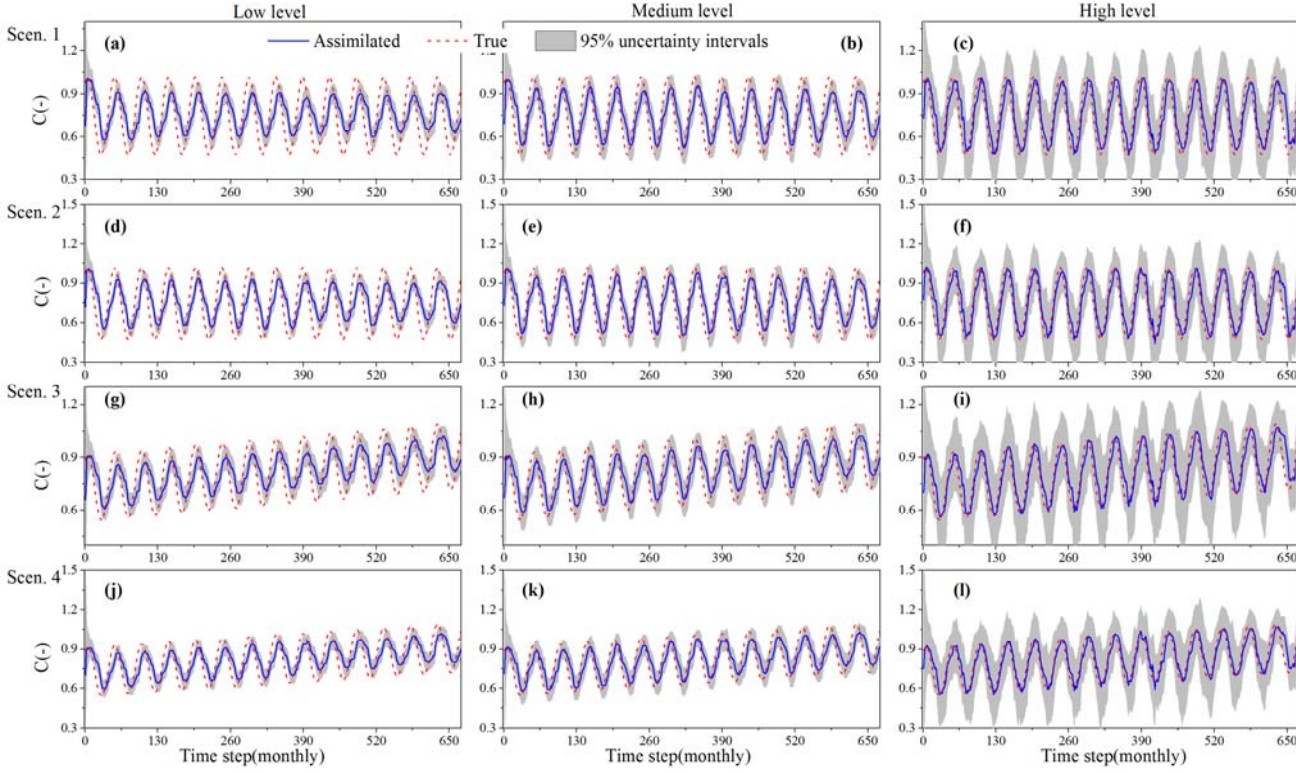


**Figure. 3.** Comparison between estimated $C$ and its true values for various parameter changes under different uncertainty levels. The grey areas represent the 95% uncertainty intervals.




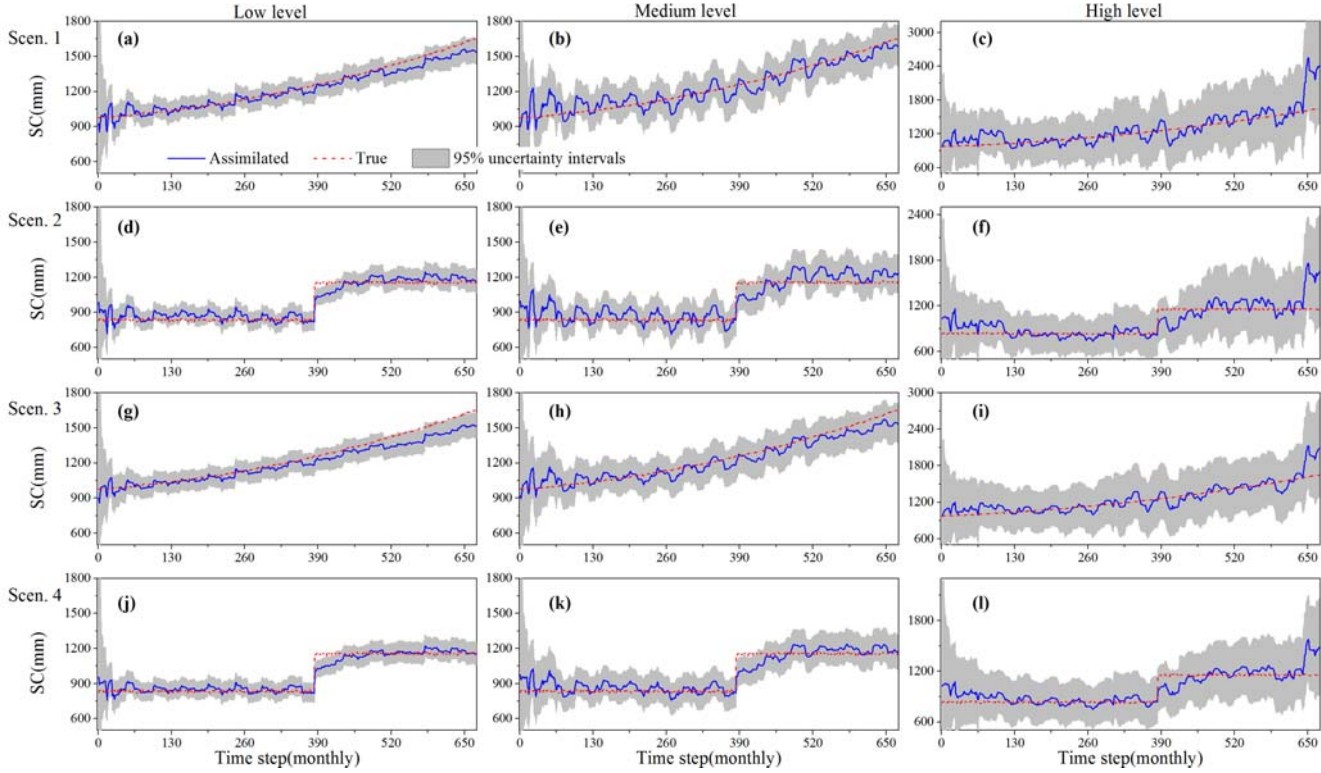


**Figure. 4.** Comparison between estimated *SC* and its true values for various parameter changes under different uncertainty levels. The grey areas represent the 95% uncertainty intervals.




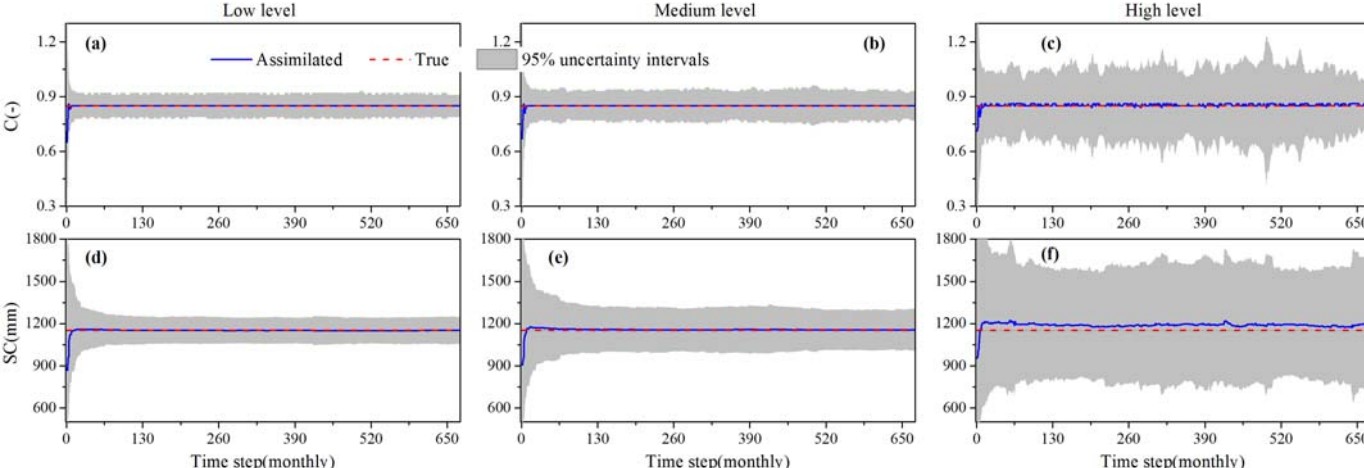


**Figure. 5.** Estimations of time-invariant *C* and *SC* under different uncertainty levels. The grey areas represent the
95% uncertainty intervals.






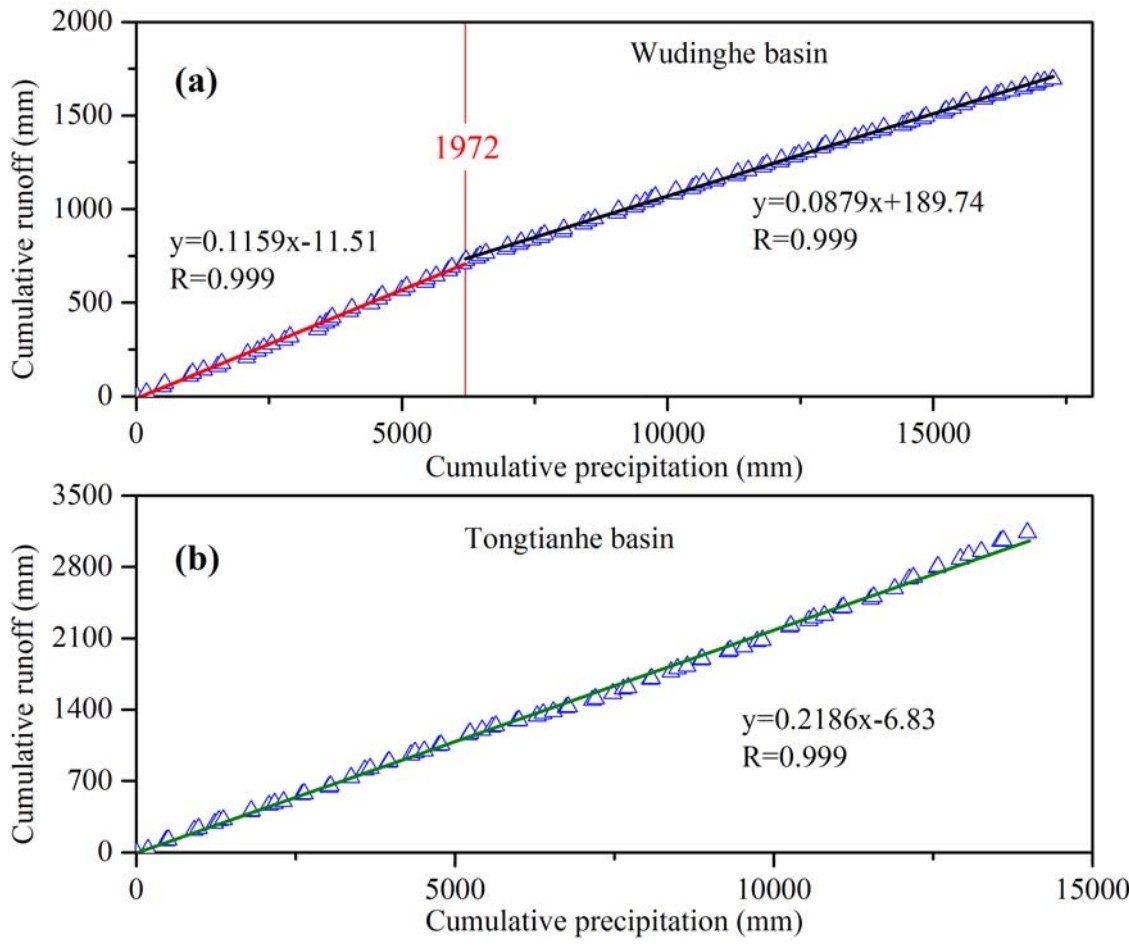


**Figure. 6.** Double mass curve between monthly runoff and precipitation for Wudinghe basin within the period of
1958-2000 (top figure) and Tongtianhe basin within the period of 1982-2013 (bottom), respectively.






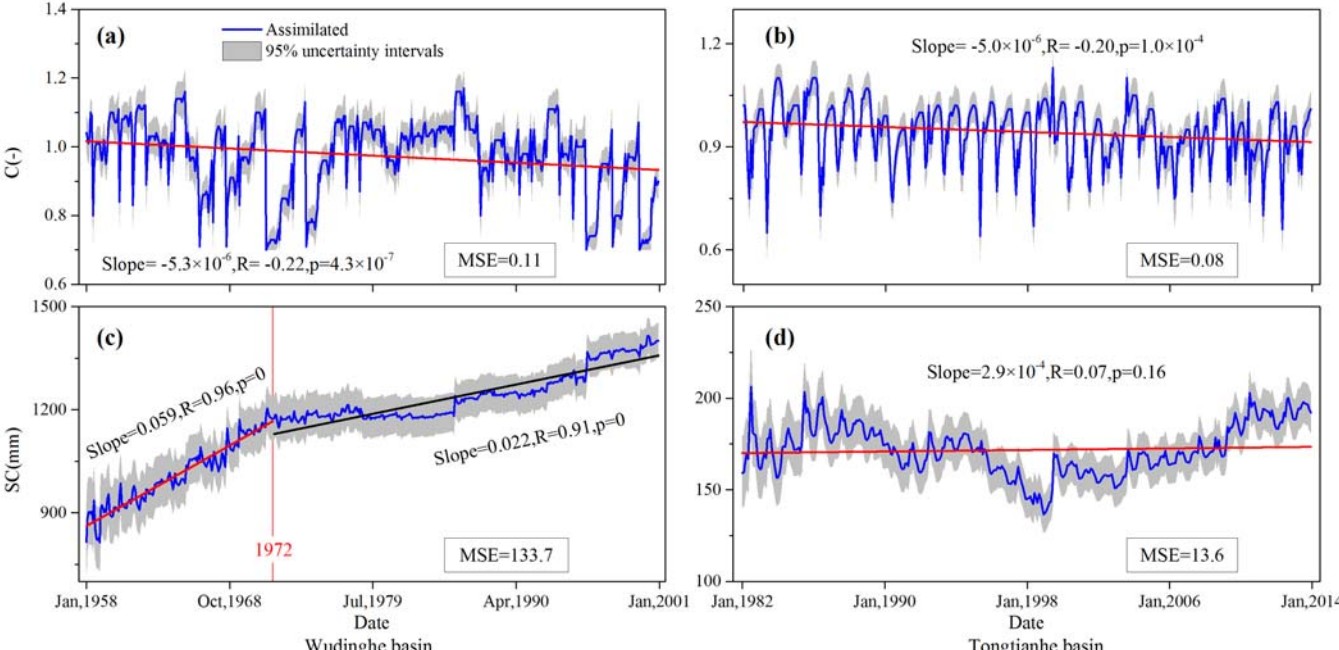


**Figure. 7.** Estimated parameter values of *C* and *SC* for (1) Wudinghe basin within the period of 1958-2000, and (2) Tongtianhe basin within the period of 1982-2013. The grey areas represent the 95% uncertainty intervals. Note that the MSE denotes the standard deviation of the estimated parameter values.




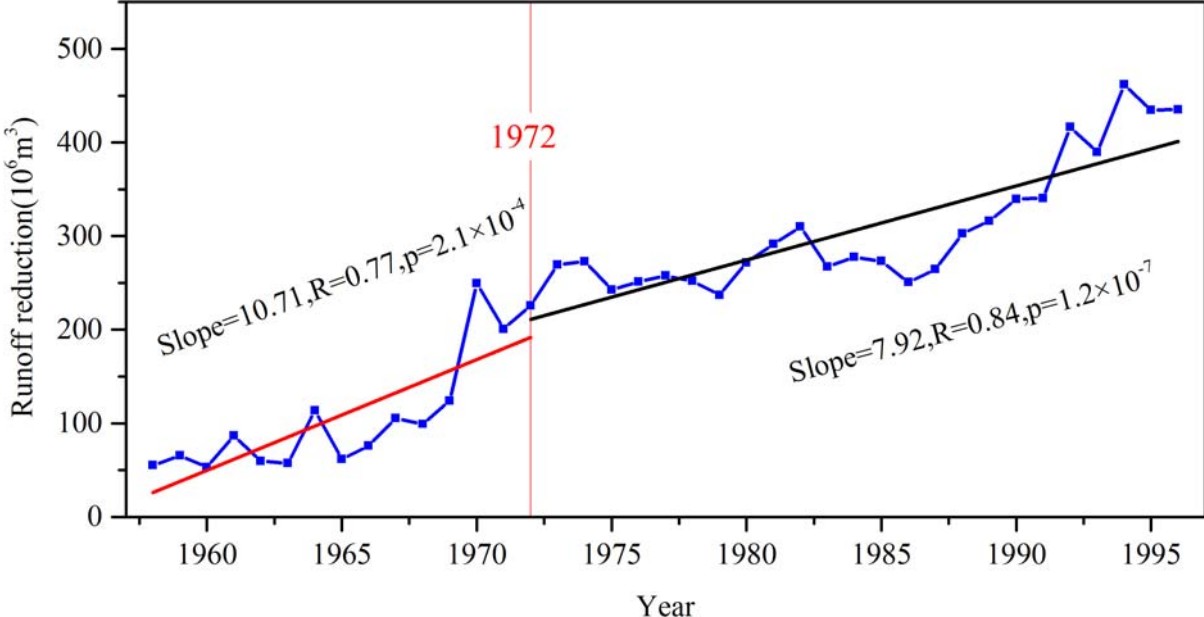


**Figure 8.** Runoff reduction in Wudinghe basin caused by all the soil and water conservation measures, i.e., land
terracing, tree and grass plantation, check dam and reservoir construction for the period of 1958- 1996. The data is
from Wang and Fan, 2003.