# Peer review of "Identification of hydrological model parameters variation using"

_Hydrology and Earth System Sciences, 2016_

## Referee Comment (RC1) · Anonymous Referee #1 · 7 Sep 2016

In this manuscript, the authors investigate whether it is possible to infer the temporal variability of certain hydrological model parameters that are often assumed to be stationary. To that end, the method of ensemble Kalman (EnKF) filter is applied, which is known for its ability to account for time-varying state variables. The authors apply their approach first to a synthetic basin with varying degrees of uncertainty and then to two different real-world basins with different temporal variability of model parameters. Their results demonstrate the overall ability of EnKF for time-variant parameter identification.

The manuscript itself is very well written. The introduction gives an adequate overview on the relevant questions and properly motivates the study. The methods section provides the reader with the necessary information on the used model, the EnKF used for the inference and the criteria used for evaluating success. The results are presented in a way that it easy to follow and understand, and the discussion provides the necessary

context for these results. The data given through figures and tables is clear, well presented and sufficient to support the conclusions drawn by the authors. Furthermore, the presented conclusions are very relevant for the Scientific Community interested in model calibration and are well suited for the scope of HESS. I have to say that I really liked the study and the way it is presented in the manuscript. I can not see any major problems and I think the authors did a fine job throughout. In conclusion, I would strongly recommend publication.

In the following, I will list a number of minor concerns that could be easily fixed to improve the manuscript even further. None of them, however, affect the overall quality.

- Page 5, Line 62: The authors present two established methods to account for time-variant parameters: windowed assimilation (dividing the calibration set into smaller subsets) and parametric assimilation (assuming a parametric model for the time dependency) and contrast this with EnKF which is an non-parametric assimilation procedure (no form of the time dependency is assumed). I wonder how their approach might fare against parametric techniques. Typically, parametric estimation techniques are superior when the true form of the dependency is known but their performance quickly decreases when this condition isn't meet. Maybe, the authors want to elaborate where they see the strengths and weaknesses of their method vis-a-vis these other approaches. This may be relevant for paramters like $C$ (the evapotranspiration paramter), where plausible paramteric models for the time dependency are possible. In fact, the authors use a paramteric model for $C$ (for simulation and not for estimation, of course) in their synthetic basin. In such a situation, a parametric estimation scheme may outperform EnKF.

- Page 6, Line 82: The authors use the term data assimilation of which EnKF is a particular implementation. The term is introduced in the introduction together with its abbreviation and never used again. If you introduce a term, it better be

important later on. If not, I would propose to skip this term and start with EnKF right away.

- Page 8, Line 17: The authors say that EnKF is based on the Monte-Carlo method. I am not sure about the wording. First, Monte Carlo is not really a method but a buzzword for virtually any method that employs a random number generator at some point. Second, the randomness is only one element of EnKF, with others being the approximation of the covariance by the sample covariance and the assumption of Gaussianity for the PDF's.

- Page 8, Line 19: The authors care to mention that EnKF is applicable to a variety of non-linear problems. I am not an expert on the issue but I alway thought that EnKF assumes a linear forward model. I know that extensions of the Kalman filter to non-linear models exist. Is that what the authors talk about? If so, it's a little bit confusing.

- The authors consistently speak of uncertainty intervals (e.g., Page 19, Line 14). What do they mean by that? Credible intervals, confidence intervals, prediction intervals or something else? In my opinion, only credible intervals represent uncertainty, so the authors should elaborate on what they mean.

- Page 13, Line 97: If the authors care to explain that $NSE = 1$ is a perfect match, they should also explain that it starts at $-\infty$. People, who do not know about the NSE, may be lead to think that it varies between $0 < NSE < 1$, which is obviously not the case. On the other hand, people who do know about the NSE don't need that information.

- The authors diverge from the established IMRaD structure by splitting the Methods part into the 'Methodology' and 'Data and study area' section. This is nothing major, but it was a little bit disorienting when I first read the manuscript.

- The manuscript appears to have been typeset with a word processor like Microsoft Word and it shows. There are several major widows and orphans throughout the manuscript (e.g., Page 4, 7, 8, 11, and 24). I guess the publishing office takes care of it in the final version, but it was a drag while reading. In particular, section headings shouldn't be left dangling on a single page (see, e.g., Page 13 and 15).

- Similarly, the line numbering was confusing. Either use continuous line numbering or start anew every page.

- Punctuation is missing throughout all equations that aren't inline. Punctuation rules should apply to both inline and non-inline equations (see, e.g., Higham, Nicholas J. (1998), Handbook of Writing for the Mathematical Sciences, SIAM, ISBN 0-89871-420-6).

- Instead of acknowledging the contribution of the reviewers (who haven't done anything at this point), the authors may want to include the data providers (e.g., the China Meteorological Data Sharing Service System).

---

## Referee Comment (RC2) · Anonymous Referee #2 · 23 Sep 2016

General evaluation:

This paper illustrates that temporally variable parameters can be estimated with EnKF. The paper can be resubmitted after major revision and I give a series of comments to be handled. The two main points are:

(1) Do the found parameter variations in the real-world case show a significant trend? Why do these parameter values fluctuate so strongly?

(2) The explanation of the apparent trend in the parameters is not convincing to me. I ask the authors to provide long-term time series of precip and potential ET, discuss the potential role of factors like increasing water use efficiency of the vegetation and increased groundwater pumping in the area. Other data sources like trends in ground-water levels would also be helpful. It should be remembered that with this very simple

hydrological model the parameters incorporate many processes and a physical interpretation is difficult.

Detailed points:

L 52: This should not give time dependent parameters and points to a problem in the model.

L62-L63: Rephrase.

L73: Add Kurtz et al. (2012, WRR) who performed a detailed study on modelling time dependent parameters for a hydrological system. Also Montzka et al. (2013, VZJ) estimated time dependent parameters.

L75: Please provide more details about this study as Vrugt et al. (2013) showed problems associated with estimating time dependent parameters.

L76: retrieve,

L80: see earlier comment.

L117: skip typical.

L119: give original references (i.e., Evensen (1994), Burgers et al. (1998)).

L137: "following" instead of "followed".

L138-L139: Why is this needed? This is normally only applied for the particle filter.

L155: give an earlier reference.

Page 10: I think it would be better to use the standard notation like overbar for an average and C for covariance matrix.

L178: What does this mean? Tuned? Trial and error? Parameters do not have physical meaning.

L186: This is however usually applied for the particle filter. Is it done here?

L228: It should be made clear and explicitly stated that these are synthetically generated parameter time series.

L275: Reformulate.

L282: What about crop/vegetation data?

L291: The estimation of parameters

L314: skip "to".

L321: "(. . .) to a certain degree"

L332: "On the other hand, the bottom panel demonstrates that (. . .)"

L341: Is the trend slope significantly different from zero? The fluctuations are so strong that this seems not so clear. These strong fluctuations should also be explained .

L349-L351: Rephrase sentence.

L357-L358: Rephrase sentence. Skip "the" and "parameter" instead of "parameters".

L380: change to: "assimilating runoff observations".

L384: skip "drawn as follows".

L405: "parameter" instead of "parameters".

Figure 8: I would expect that in the long-term the water balance should be zero and if precipitation does not decrease, why would runoff reduce? Please plot in the paper also long term time series of yearly precipitation and potential evapotranspiration. Is it possible that ET reduced in relation to other factors and that the relation between actual ET, potential ET and precipitation was related to a $CO_2$-induced change in water use efficiency of the plants? Were groundwater abstractions increased in this area?

---

## Author Comment (AC1) · 13 Oct 2016

**Response to Anonymous Referee #1**

**(1) In this manuscript, the authors investigate whether it is possible to infer the temporal variability of certain hydrological model parameters that are often assumed to be stationary. To that end, the method of ensemble Kalman (EnKF) filter is applied, which is known for its ability to account for time-varying state variables. The authors apply their approach first to a synthetic basin with varying degrees of uncertainty and then to two different real-world basins with different temporal variability of model parameters. Their results demonstrate the overall ability of EnKF for time-variant parameter identification.**

**The manuscript itself is very well written. The introduction gives an adequate overview on the relevant questions and properly motivates the study. The methods section provides the reader with the necessary information on the used model, the EnKF used for the inference and the criteria used for evaluating success. The results are presented in a way that it easy to follow and understand, and the discussion provides the necessary context for these results. The data given through figures and tables is clear, well presented and sufficient to support the conclusions drawn by the authors. Furthermore, the presented conclusions are very relevant for the Scientific Community interested in model calibration and are well suited for the scope of HESS. I have to say that I really liked the study and the way it is presented in the manuscript. I cannot see any major problems and I think the authors did a fine job throughout. In conclusion, I would strongly recommend publication.**

**Reply:**

We thank the reviewer for the positive summary and helpful comments.

**(2) Page 5, Line 62: The authors present two established methods to account for time-variant parameters: windowed assimilation (dividing the calibration set into smaller subsets) and parametric assimilation (assuming a parametric model for the time dependency) and contrast this with EnKF which is a non-parametric**

assimilation procedure (no form of the time dependency is assumed). I wonder how their approach might fare against parametric techniques. Typically, parametric estimation techniques are superior when the true form of the dependency is known but their performance quickly decreases when this condition isn't meet. Maybe, the authors want to elaborate where they see the strengths and weaknesses of their method vis-a-vis these other approaches. This may be relevant for parameters like C (the evapotranspiration parameter), where plausible parametric models for the time dependency are possible. In fact, the authors use a parametric model for C (for simulation and not for estimation, of course) in their synthetic basin. In such a situation, a parametric estimation scheme may outperform EnKF.

**Reply:**

As the reviewer mentioned, the performance of the parametric estimation is significantly affected by the catchment conditions (e.g., climate and vegetation), and it is difficult to obtain the true form of the parameter function. We agree with the reviewer that a parametric estimation scheme may have a better performance if the true parameter function can be obtained. Even though the EnKF-based estimation cannot perfectly match the time-variant values of the parameters, it can successfully capture the temporal variations of the parameters based on the results from the synthetic experiment. The results from the two case studies show that the estimated time series of the parameters can be linked to the variations of the catchment characteristics, illustrating the good performance of the proposed method. One of the advantages for estimating the time-variant parameters using the EnKF is that it can conduct real time updating for the parameters based on the observations, providing time series of parameter values without assuming the parameter functions or sub-dividing the calibration set.

**(3) Page 6, Line 82: The authors use the term data assimilation of which EnKF is a particular implementation. The term is introduced in the introduction together with its abbreviation and never used again. If you introduce a term, it better be**

**important later on. If not, I would propose to skip this term and start with EnKF right away.**

**Reply:**

Thank you. The data assimilation methods applied in hydrology include EnKF and others such as Particle-DREAM. EnKF is a typical data assimilation method. We have revised the aim for clarification (Page 6, Line 84-85).

"The aim of this study is to assess the capability of the EnKF to identify the temporal variations of the model parameters for a monthly water balance model."

**(4) Page 8, Line 17: The authors say that EnKF is based on the Monte-Carlo method. I am not sure about the wording. First, Monte Carlo is not really a method but a buzzword for virtually any method that employs a random number generator at some point. Second, the randomness is only one element of EnKF, with others being the approximation of the covariance by the sample covariance and the assumption of Gaussianity for the PDF's.**

**Reply:**

Thank you. We agree with the comment. The Monte Carlo is not really a method, and the EnKF is not only based upon the Monte Carlo but also the Kalman filter formulation. The wording has been modified in the revised manuscript (Page 8, Line 119-121).

"As a sequential data assimilation technique, EnKF is based on the Monte Carlo and the Kalman filter formulation to produce an ensemble of state simulations for updating the state variables and their covariance matrix, conditioned on a series of observations (Evensen 1994; Burgers et al., 1998; Moradkhani et al., 2005; Shi et al., 2014)."

**(5) Page 8, Line 19: The authors care to mention that EnKF is applicable to a variety of non-linear problems. I am not an expert on the issue but I alway thought that EnKF assumes a linear forward model. I know that extensions of the Kalman filter to non-linear models exist. Is that what the authors talk about?**

**If so, it's a little bit confusing.**

**Reply:**

Thank you. The standard Kalman filter (KF), which is a data assimilation technique for linear systems, has been modified to the Extended Kalman filter (EKF) for nonlinear problems. EKF is used for linear approximation and has limits in estimation stability when the nonlinearity degree increases in the system. Ensemble Kalman filter (EnKF) uses statistical distributions to represent uncertainties of model and observation errors and to produce ensembles for updating state and parameter variables. EnKF has been used for a variety of nonlinear problems (Evensen, 2003; Weerts and El Serafy, 2006), especially for the estimation of model states and parameters (Moradkhani et al., 2005; Wang et al., 2009; Xie and Zhang, 2010; Xie and Zhang, 2013; Samuel et al., 2014). Therefore, we use EnKF to identify the temporal variations of model parameters in this study since the hydrologic model is nonlinear.

**(6) The authors consistently speak of uncertainty intervals (e.g., Page 19, Line 14). What do they mean by that? Credible intervals, confidence intervals, prediction intervals or something else? In my opinion, only credible intervals represent uncertainty, so the authors should elaborate on what they mean.**

**Reply:**

Thank you. The uncertainty intervals used in this study are prediction intervals, which are obtained from the updated ensembles of the model parameters (Vrugt et al., 2013). It has been clarified in the revised manuscript (Page 19, Line 311-313).

"The grey areas represent the 95% prediction uncertainty intervals, which reduce quickly and approach a stable spread."

**(7) Page 13, Line 97: If the authors care to explain that NSE=1 is a perfect match, they should also explain that it starts at -∞. People, who do not know about the NSE, may be lead to think that it varies between 0<NSE<1, which is obviously not the case. On the other hand, people who do know about the NSE don't need**

**that information.**

**Reply:**

Thank you. The explanations are added to clarify the meanings of *NSE* values (Page 12-13, Line 195-198).

"It ranges from $-\infty$ to 1. A *NSE* value of 1 means a perfect match of simulated runoff to the observations, while a value of 0 means the model simulations are the same as the mean value of the runoff observations; and negative *NSE* values indicate that the mean observed runoff is better than the model simulations."

**(8) The authors diverge from the established IMRaD structure by splitting the Methods part into the 'Methodology' and 'Data and study area' section. This is nothing major, but it was a little bit disorienting when I first read the manuscript.**

**Reply:**

Thank you. The "Data and study area" part includes a synthetic experiment and two case studies. Therefore, we split the Methods into two parts.

**(9) The manuscript appears to have been typeset with a word processor like Microsoft Word and it shows. There are several major widows and orphans throughout the manuscript (e.g., Page 4, 7, 8, 11, and 24). I guess the publishing office takes care of it in the final version, but it was a drag while reading. In particular, section headings shouldn't be left dangling on a single page (see, e.g., Page 13 and 15).**

**Reply:**

Thanks. The widows and orphans have been adjusted in the revised manuscript.

**(10) Similarly, the line numbering was confusing. Either use continuous line numbering or start a new every page.**

**Reply:**

Thanks. We are not sure if the reviewer got the right pdf version of the manuscript,

but the line numbering in the file "hess-2016-370.pdf" is continuous (Line 1 to 629) from Page 1 to 42 after double checked.

**(11) Punctuation is missing throughout all equations that aren't inline. Punctuation rules should apply to both inline and non-inline equations (see, e.g., Higham, Nicholas J. (1998), Handbook of Writing for the Mathematical Sciences, SIAM, ISBN 0-89871-420-6).**

**Reply:**

Thanks. The writing of symbols and equations is checked and revised. Punctuation is added for all the equations (Page 9, Line 131; Page 7-13).

**(12) Instead of acknowledging the contribution of the reviewers (who haven't done anything at this point), the authors may want to include the data providers (e.g., the China Meteorological Data Sharing Service System).**

**Reply:**

Thanks. The acknowledgement to data provider has been added (Page 25, Line 415-416).

"The authors thank the China Meteorological Data Sharing Service System for providing a part of the data used in this study."

---

## Author Comment (AC2) · 13 Oct 2016

**Response to Anonymous Referee #2**

**(1) This paper illustrates that temporally variable parameters can be estimated with EnKF. The paper can be resubmitted after major revision and I give a series of comments to be handled. The two main points are:**

**1) Do the found parameter variations in the real-world case show a significant trend? Why do these parameter values fluctuate so strongly?**

**Reply:**

(a) The estimates of parameter *SC* from Wudinghe basin (Fig. 7c) show a significant increasing trend (p-value=0); while the estimated *SC* from Tongtianhe basin has no obvious trend since the correlation coefficient has an insignificance level (p-value=0.16). For parameter *C*, the results show that the estimates have no significant temporal patterns because the slopes for the trend line are near zero and the standard deviations are relatively small for the two basins (Fig. 7(a) and (b)).

(b) The fluctuations are mostly caused by the modeling and observation uncertainties (Shi et al., 2014; Meng et al., 2016). To reflect these uncertainties, the standard deviations of observations and parameters are set, respectively, shown in Table 3 and Section 2.2 (Page 11-12, Line 177-185). The results from Figures 3 to 5 show that stronger fluctuations appear when higher standard deviations are set. This is also illustrated in Page 11, Line 174-177. The set of the standard deviations is based on trial and error and the related previous studies (Moradkhani et al., 2005; Wang et al., 2009; Xie and Zhang, 2010; Nie et al., 2011; Lü et al., 2013; Samuel et al., 2014).

**2) The explanation of the apparent trend in the parameters is not convincing to me. I ask the authors to provide long-term time series of precipitation and potential ET, discuss the potential role of factors like increasing water use efficiency of the vegetation and increased groundwater pumping in the area. Other data sources like trends in groundwater levels would also be helpful. It should be remembered that with this very simple hydrological model the parameters incorporate many processes and a physical interpretation is difficult.**

**Reply:**

As the reviewer mentioned, besides the soil and water conservation measures, other potential factors such as precipitation alteration and groundwater pumping can also affect the runoff reduction (Wang and Cai.: *Detecting human interferences to low flows through base flow recession analysis*, Water Resour. Res., 2009).

The data used to illustrate the trends of parameter *SC* from Wudinghe basin is from a program report by Wang and Fan (2003) that specifically study the water and sediment changes resulted from the different factors including precipitation and human activities. This study showed that the runoff reduction are mainly caused by human activities, which were the soil and water conservation measures, i.e., land terracing, tree and grass plantation, check dam and reservoir construction. All the possible human activities have been considered in this study and the groundwater abstractions is negligible in Wudinghe basin.

The monthly water balance model used in this study is a simple conceptual model with only two parameters, i.e., evapotranspiration parameter and catchment water storage capacity. These two parameters have clear physical means. As the reviewer mentioned, these parameters are affected by multiple factors. In this manuscript, we use two study areas with different catchment characteristics to evaluate the proposed method.

The long-term time series of precipitation and potential ET have been added in the revised manuscript (Page 21, Line 346-351). We agree that other data sources like the groundwater level series would also be helpful. Unfortunately, these data are not available.

"Fig. 8 shows the long-term time series of precipitation and potential evaporation in Wudinghe basin, and the runoff reduction caused by all the soil and water conservation measures, i.e., land terracing, tree and grass plantation, check dam and reservoir construction. Fig. 8(a) shows that the yearly potential evaporation has no significant trend; while both yearly precipitation and runoff have a decreasing trend, and the trend of the yearly precipitation has a higher slope. Runoff decreases significantly while precipitation changes slightly and potential evaporation has no

trend, indicating that the actual evaporation increases significantly due to impacts of human activities, i.e., the soil and water conservation measures."

[Figure]

**Figure 1.** (a) Yearly precipitation, potential evaporation and runoff in Wudinghe basin during the period of 1958-2000; (b) Runoff reduction in Wudinghe basin caused by all the soil and water conservation measures, i.e., land terracing, tree and grass plantation, check dam and reservoir construction for the period of 1958- 1996. Note that the data is from Wang and Fan (2003) and is only available from 1956 to 1996.

**(2) L52: This should not give time dependent parameters and points to a problem in the model.**

**Reply:**

Thanks. This sentence has been modified (Page 4, Line 50-51).

"Therefore, assuming time-invariant model parameters may be unrealistic, especially for catchments with nonstationary catchment characteristics."

**(3) L62-L63: Rephrase.**

**Reply:**

Thanks. This sentence has been rephrased (Page 5, Line 61-63).

"(1) Available historical record is divided into consecutive subsets, and parameters are calibrated separately for each subset using an optimization algorithm (Merz et al., 2011; Thirel et al., 2015);"

**(4) L73: Add Kurtz et al. (2012, WRR) who performed a detailed study on modelling time dependent parameters for a hydrological system. Also Montzka et al. (2013, VZJ) estimated time dependent parameters.**

**Reply:**

Thanks. References are added in the revised manuscript (Page 6, Line 73).

**(5) L75: Please provide more details about this study as Vrugt et al. (2013) showed problems associated with estimating time dependent parameters.**

**Reply:**

Thanks. More details about the paper by Vrugt et al. (2013) have been added (Page 6, Line 74-77).

"Vrugt et al. (2013) proposed two types of Particle-DREAM method, i.e., Particle-DREAM for time-variant parameters and time-invariant parameters, to track the evolving target distribution of HyMOD parameters, while both the results were approximately similar and statistically coherent since only three years of data were used."

**(6) L76: retrieve.**

**Reply:**

Thanks. It has been corrected.

**(7) L80: see earlier comment.**

**Reply:**

Thanks. This sentence has been modified (Page 6, Line 80-82).

"Little attention has been paid to the identification of time-variant model parameters and the interpretation of their temporal variations based on catchment characteristics."

**(8) L117: skip typical.**

**Reply:**

Thanks. It has been modified.

**(9) L119: give original references (i.e., Evensen (1994), Burgers et al. (1998)).**

**Reply:**

The original references have been added in the revised manuscript (Page 8, Line 121).

**(10) L137: "following" instead of "followed".**

**Reply:**

Thanks. It has been revised.

**(11) L138-L139: Why is this needed? This is normally only applied for the particle filter.**

**Reply:**

The simple random walk process is used to represent the propagation of parameters (Wang et al., 2009), i.e., small random disturbances are added to the parameter member between time steps as in equation (5).

**(12) L155: give an earlier reference.**

**Reply:**

Earlier reference has been added (Page 10, Line 157).

**(13) Page 10: I think it would be better to use the standard notation like overbar for an average and C for covariance matrix.**

**Reply:**

Thanks. The notation for average has been changed, while that for covariance matrix is kept since $C$ is used to denote the evapotranspiration parameter (Page 11, Line 164).

**(14) L178: What does this mean? Tuned? Trial and error? Parameters do not**

**have physical meaning.**

**Reply:**

It is the standard deviations of the two parameters. To reflect these uncertainties, the standard deviations of observations and parameters are set, respectively. The set of the standard deviations is based on a trial and error method and the related previous studies (Moradkhani et al., 2005; Wang et al., 2009; Xie and Zhang, 2010; Nie et al., 2011; Lü et al., 2013; Samuel et al., 2014).

**(15) L186: This is however usually applied for the particle filter. Is it done here?**

**Reply:**

No, the variable variance multiplier is not used here. The description has been deleted.

**(16) L228: It should be made clear and explicitly stated that these are synthetically generated parameter time series.**

**Reply:**

Thanks. It has been modified (Page 14, Line 228).

"Time series of model parameters are synthetically generated, including the time-variant parameters and the constant parameters."

**(17) L275: Reformulate.**

**Reply:**

Thanks. The sentence has been rephrased (Page 17, Line 273-274).

"The Tongtianhe basin is rarely affected by human activities owing to the water source protection guidelines conducted by the government."

**(18) L282: What about crop/vegetation data?**

**Reply:**

The modeling time scale of this study is monthly, the corresponded crop and vegetation (e.g., monthly or yearly) data are unavailable in the study area.

**(19) L291: The estimation of parameters.**

**Reply:**

Thanks. The words have been revised.

**(20) L314: skip "to".**

**Reply:**

Thanks. It has been deleted.

**(21) L321: "(...) to a certain degree"**

**Reply:**

Thanks. It has been modified.

**(22) L332: "On the other hand, the bottom panel demonstrates that (...)"**

**Reply:**

Thanks. This sentence has been revised.

**(23) L341: Is the trend slope significantly different from zero? The fluctuations are so strong that this seems not so clear. These strong fluctuations should also be explained.**

**Reply:**

Yes, the trend slopes in Fig. 7(c) are significantly different from zero since both the p-values of the trend lines are equal to zero. The values are small because the date values are used as independent variable. The fluctuations are caused by the modeling and observation uncertainties (Shi et al., 2014; Meng et al., 2016). To reflect these uncertainties, the standard deviations of observations and parameters are set respectively. Stronger fluctuations appear when higher standard deviations are set. This is illustrated in Page 11, Line 174-177.

**(24) L349-L351: Rephrase sentence.**

**Reply:**

Thanks. The sentence has been rephrased (Page 21, Line 353-355). The sentence "The runoff reduction data is available from 1956 to 1996 (Wang and Fan, 2003)" has been moved to the caption of Fig. 8.

**(25) L357-L358: Rephrase sentence. Skip "the" and "parameter" instead of "parameters".**

**Reply:**

Thanks. The sentence has been revised (Page 22, Line 361-362).

"However, it can be treated as time-variant parameter since temporal variations exist in the estimated $C$ series."

**(26) L380: change to: "assimilating runoff observations".**

**Reply:**

Thanks. It has been modified.

**(27) L384: skip "drawn as follows".**

**Reply:**

Thanks. These words have been deleted.

**(28) L405: "parameter" instead of "parameters".**

**Reply:**

Thanks. It has been modified in the revised manuscript.

**(29) Figure 8: I would expect that in the long-term the water balance should be zero and if precipitation does not decrease, why would runoff reduce? Please plot in the paper also long term time series of yearly precipitation and potential evapotranspiration. Is it possible that ET reduced in relation to other factors and that the relation between actual ET, potential ET and precipitation was related to a CO2-induced change in water use efficiency of the plants? Were groundwater abstractions increased in this area?**

**Reply:**

As the reviewer mentioned, besides the soil and water conservation measures, other factors such as precipitation and groundwater pumping can also affect the runoff reduction. While the data used to illustrate the trends of parameter *SC* is from a research report by Wang and Fan (2003) that specifically studied the water and sediment changes resulted from the different factors including precipitation and human activities (i.e., land terracing, tree and grass plantation, check dam and reservoir construction). The data used in Figure 8 is runoff reduction only caused by human activities, i.e., the soil and water conservation measures.

In the study by Wang and Fan (2003), the trends of the yearly precipitation and runoff have been analyzed, and an empirical yearly runoff model has been built to compute the runoff changes caused by precipitation and human activities, respectively. **Figure R1** shows that the yearly potential evaporation has no significant trend; while both yearly precipitation and runoff have a decreasing trend, and the trend of the yearly precipitation has a higher slope. Runoff decreases significantly while precipitation does not change much and potential evaporation has no trend, indicating that the actual evaporation increases significantly due to impacts of human activities, i.e., the soil and water conservation measures. All the possible human activities have been considered in this study and the groundwater abstractions is negligible.

The long-term time series of yearly precipitation, potential evapotranspiration and runoff have been added in the revised manuscript (Page 43, Line 641-646).

[Figure]

**Figure R1.**  Yearly precipitation, potential evaporation and runoff in Wudinghe basin during the period of 1958-2000.